

# Screening of a natural compound library identifies emodin, a natural compound from *Rheum palmatum* Linn that inhibits DPP4

Zhaokai Wang[1,2], Longhe Yang[2], Hu Fan[2], Peng Wu[2], Fang Zhang[2], Chao Zhang[2], Wenjie Liu[3] and Min Li[1]

[1] College of Life Sciences, Fujian Normal University, Fuzhou, P. R. China
[2] Engineering Research Center of Marine Biological Resource Comprehensive Utilization, Third Institute of Oceanography, State Oceanic Administration, Xiamen, P. R. China
[3] Fujian Provincial Key Laboratory of Innovative Drug Target Research, School of Pharmaceutical Sciences, Xiamen, P. R. China

## ABSTRACT

Historically, Chinese herbal medicines have been widely used in the treatment of hyperglycemia, but the mechanisms underlying their effectiveness remain largely unknown. Here, we screened a compound library primarily comprised of natural compounds extracted from herbs and marine organisms. The results showed that emodin, a natural compound from *Rheum palmatum* Linn, inhibited DPP4 activity with an in vitro $IC_{50}$ of 5.76 $\mu$M without inhibiting either DPP8 or DPP9. A docking model revealed that emodin binds to DPP4 protein through Glu205 and Glu206, although with low affinity. Moreover, emodin treatment (3, 10 and 30 mg/kg, P.O.) in mice decreased plasma DPP4 activity in a dose-dependent manner. Our study suggests that emodin inhibits DPP4 activity and may represent a novel therapeutic for the treatment of type 2 diabetes.

## INTRODUCTION

Type 2 diabetes mellitus (T2DM) is a metabolic disease associated with insulin resistance and pancreatic $\beta$-cell failure (*Defronzo, 2009*). Therefore, enhancing pancreatic insulin secretion while protecting pancreatic $\beta$-cells represents a promising therapeutic approach for the treatment of type 2 diabetes. Glucagon-like peptide 1 (GLP-1) is one of the incretin hormones released from cells in the gastrointestinal tract in response to nutrient absorption. Incretin hormones, especially GLP-1, regulate post-prandial insulin secretion by inhibiting glucagon release and stimulating insulin biosynthesis and secretion (*Baggio & Drucker, 2007*). In T2DM patients, GLP-1 is critical for glucose homeostasis (*Mulvihill & Drucker, 2014*).

Dipeptidyl peptidase 4 (DPP4), which was first identified by Hopsu-Havuand Glenner, rapidly degrades the active form of GLP-1 ($GLP-1_{7-36}$) to inactive $GLP-1_{9-36}$ within minutes *in vivo* (*Hopsu-Havu & Glenner, 1966*; *Mulvihill & Drucker, 2014*). DPP4 is commonly

Corresponding authors
Wenjie Liu, wjliu@xmu.edu.cn
Min Li, mli@fjnu.edu.cn

expressed as two forms: a membrane-associated and soluble circulating protein and a cleaved protein containing either analanine or proline at position 2 (*Lambeir et al., 2003*). Therefore, a DPP4 inhibitor could potentially increase the effect of intact GLP-1, thus prolonging its anti-diabetic effects (*Smith et al., 2014*).

Although several DPP4 inhibitors such as sitagliptin (MK-0431) (*Kim et al., 2005*), vildagliptin (LAF-237) (*Villhauer et al., 2003*), saxagliptin (BMS-477118) (*Augeri et al., 2005*), alogliptin (SYR-322) (*Feng et al., 2007*) and linagliptin (BI-1356) (*Eckhardt et al., 2007*) have been approved for the treatment of T2DM, few natural compounds have been reported to exert DPP4 inhibitory activity (*Geng et al., 2013*).

Traditional Chinese medicine (TCM) has been used in the clinical treatment of diabetes and related complications for centuries (*Wang & Chiang, 2012*; *Xie & Du, 2011*). *Radix Astragali* (*Wang et al., 2009*) and *Radix Rehmanniae* (*Huang et al., 2010*) are TCMs with both hypoglycemic and anti-inflammatory activities as reviewed by *Xie & Du (2011)* and *Liu et al. (2002)*. However, the underlying mechanisms of the effective components are largely unknown because of the poor characterization of Chinese medicine. Herein, we screened a small library of natural products from Chinese herbal medicines and marine organisms to identify new molecules that inhibit DPP4 activity. In our research, we discovered that emodin from the herb *Rheum palmatum* Linn inhibited DPP4 activity with an $IC_{50}$ of 5.76 µM without inhibiting of either DPP8 or DPP9. Moreover, oral administration of emodin decreased DPP4 activity in a dose-dependent manner in mice.

## MATERIALS AND METHODS

### Materials

The natural product library derived from Chinese herbs was purchased from Selleck Chemicals (Cat# L1400, Shanghai, China). Marine derived compounds were isolated and purified from marine organisms in our lab.

### DPP4 activity assay

The DPP4 screening assay was conducted using a DPP4 inhibitor screening assay kit (Cayman Chemical, Ann Arbor, MI, USA), following the manufacturer's protocol. Briefly, 30 µl of diluted assay buffer, 10 µl of diluted DPP4, and 10 µl of inhibitor were added to a 96-well plate. The reaction was initiated by adding 50 µl of diluted substrate solution to all of the wells, and this was followed by incubation with a plate cover at 37 °C for 30 min. After incubation, the fluorescence was read using an excitation wavelength of 360 nm and an emission wavelength of 460 nm.

### DPP8 activity assay

A DPP8 assay kit was purchased from BPS Bioscience (Cat# 800208), and the assay protocol was followed to test for inhibitory activity on DPP8. Briefly, DPP substrate 1 was diluted to make a 100 µM stock solution, and DPP8 protein was diluted in DPP assay buffer to 2 ng/µl (20 ng/reaction). For the tested compounds, 10 µl of diluted DPP8 protein, 5 µl of diluted DPP substrate 1, 84 µl of DPP assay buffer and 1 µl of inhibitor were added

into the assay system for a total volume of 100 μl. The reaction mixtures were prepared in duplicate on a 96-well plate and incubated at room temperature for 10 min. The plate was read on an Envision plate reader (Perkin-Elmer, Waltham, MA, USA) capable of excitation at 365 nm and emission detection at 460 nm.

## DPP9 activity assay

A DPP9 assay kit was purchased from BPS Bioscience (Cat# 800209), and the assay protocol was followed to test for inhibitory activity against DPP9. Briefly, DPP substrate 1 was diluted to make a 100 μM stock solution, and DPP9 protein was diluted in DPP assay buffer to 2 ng/μl (20 ng/reaction). For the tested compounds, 10 μl of diluted DPP9 protein, 5 μl of diluted DPP substrate 1, 84 μl of DPP assay buffer and 1 μl of inhibitor were added into the assay system for a total volume of 100 μl. The reaction mixtures were prepared in duplicate on a 96-well plate and incubated at room temperature for 10 min. The plate was read on an Envision plate reader (Perkin-Elmer, Waltham, MA, USA) capable of excitation at 365 nm and emission detection at 460 nm.

## Docking assay

Docking of compounds to the DPP4 active site was modeled using the Glide package. The 3-dimensional model of DPP4 (PDB code: 2ONC) was used in the molecular modeling experiment (*Huang et al., 2010*). Compounds were docked onto the DPP4 binding site at a position in which either the substrate or small molecule inhibitors were fitted into the active pocket. Bond formation between the compound and the DPP4 active site was dynamically simulated.

## Dialysis assay

Dialysis assay was performed using Slide-A-Lyzer Dialysis Cassettes (Pierce, Shanghai, China). Briefly, 2 mg DPP4 protein was incubated with emodin or dimethyl sulfoxide (DMSO) in 4 ml diluted assay buffer for 10 min at 37 °C. Mixed reaction solution was loaded onto a dialysis cartridge using a syringe and incubated at 4 °C for 8 h. The samples were removed from Dialysis Cassettes by syringes for DPP4 assay.

## Animal study

Balb/c mice (male, six weeks) and ob/ob (-/-) mice (male, six weeks) were purchased from the Shanghai SLAC Laboratory Animal Co. Ltd. (Shanghai, China) and maintained in an air-conditioned room at 20–25 °C under a 12 h dark/light cycle and fed certified standard chow and tap water ad libitum. Experiments were conducted in compliance with the Guide for the Care and Use of Laboratory Animals. Mice were orally administered with emodin (3, 10, or 30 mg /kg) and had their blood collected at 0.5, 1, 2, and 4 h after emodin treatment. The samples were subjected to plasma isolation immediately after collection. Plasma samples were tested for DPP4 activity with a DPP4-Glo assay kit (Promega, Beijing, China) according to the manufacturer's protocol. The experimental protocol was approved by Animal Care and Use Committee of Xiamen University(XM2015030514).

## Data analysis

Results are presented as the mean $\pm$ standard error (SEM). Differences between the groups were analyzed using multiple variances (one-way ANOVA or two-way ANOVA) followed by Bonferroni's test, with GraphPad Prism 5 software (GraphPad Software, San Diego, CA, USA). Differences were considered to be statistically significant at $p < 0.05$.

## RESULTS AND DISCUSSION

### Emodin inhibits DPP4 activity *in vitro*

To screen for novel DPP4 inhibitors from natural compounds, we established a natural compound library comprising 155 naturally derived compounds, in which 131 were isolated and purified from Chinese herbal medicines, and 24 were from marine organisms. DPP4 screening was first conducted on these 155 natural compounds by following a DPP4 screening assay kit protocol. All compounds (10 $\mu$M) were screened for DPP4 inhibitory activity. The results suggested that emodin showed greater than 50% inhibition in the DPP4 activity assay at 10 $\mu$M. Two other compounds were ruled out because of auto-fluorescence (Fig. 1A).

To further validate this finding, a dose response experiment was performed to test the inhibitory activity of emodin on DPP4. Emodin was shown to inhibit DPP4 activity *in vitro* with an $IC_{50}$ of 5.76 $\mu$M and Ki of 0.85 (Fig. 1B). The DPP4 antagonist sitagliptin was used as a positive control, which showed an $IC_{50}$ of 21.78 nM (Fig. 1C), a value similar to those from previous reports (*Kim et al., 2005*)

Considering that emodin is an anthraquinone, and many naturally occurring anthraquinones have been identified as having anti-diabetes activity (*Chen et al., 2015*; *Lin et al., 2015*; *Ramos-Zavala et al., 2011*; *Wu et al., 2014*), we wondered whether this class of compounds such as Aloe-emodin, rheochrysidin, chrysophanol and rhein, might inhibit DPP4 activity.

### Anthraquinone compounds inhibit DPP4 activity but not DPP8 or DPP9 activity

We further investigated a series of anthraquinone compounds in the DPP4 activity assay to identify potent DPP4 inhibitors in this class. Aloe-emodin, rheochrysidin, chrysophanol and rhein were tested by using the same DPP4 assay format (Table 1). The $IC_{50}$ of each compound is listed, with aloe-emodin showing an $IC_{50}$ of 16.02 $\mu$M and rhein showing an $IC_{50}$ of 23.06 $\mu$M. The $IC_{50}$ values of rheochrysidin and chrysophanol were greater than 100 $\mu$M (Table 1). These results showed that emodin was the most effective anthraquinone in inhibiting DPP4 activity. In addition, the Ki and binding energy for these compounds have also been listed in Table 1. Because of the high similarity between DPP4 and DPP8/9 and the reported toxicity of DPP8 or DPP9 inhibition in animal studies (*Lankas et al., 2005*), we tested the anthraquinone compounds in DPP8 and DPP9 activity assays. All of the compounds were tested at 100 $\mu$M, and none showed activity against either DPP8 or DPP9. Rhein showed a very weak activity on DPP8, with an IC50 greater than 100 $\mu$M. The biological function of rhein on DPP8 is minimal compared to other DPP8 inhibitors.

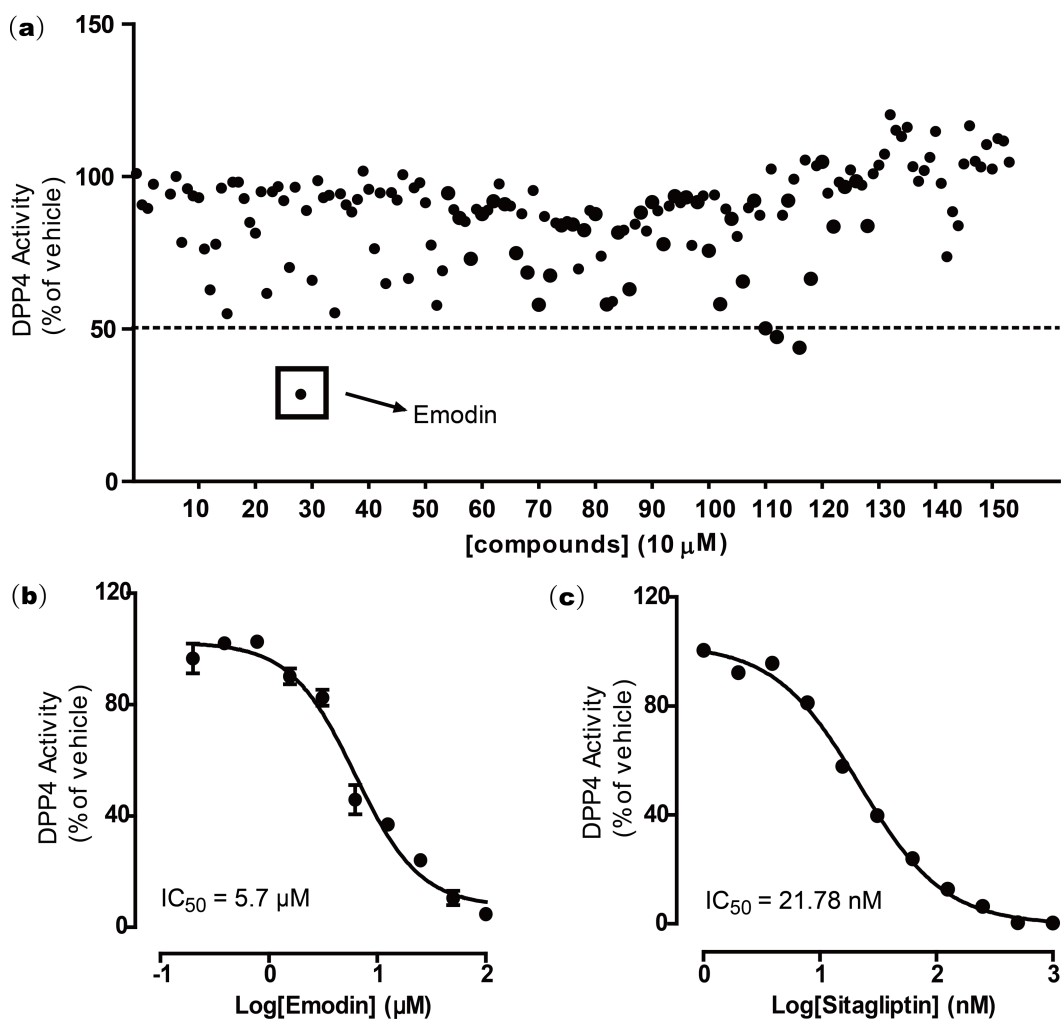

**Figure 1** Emodin was found to inhibit DPP4 activity after screening a natural compound library.

This result suggests that emodin is a relatively selective inhibitor against DPP4 (Fig. 2). To confirm this finding, we conducted a molecular docking assay.

## Emodin binds to Glu205 and Glu206 of DPP4 protein in a docking model

The active site of DPP4 consists of Arg125, Glu205, Glu206, Typ547, Trp629, Tyr666, and His740 according to the crystal structure template of DPP4 with a small molecular inhibitor (PDB code: 2ONC) (*Feng et al., 2007*). Our docking model revealed that the negatively charged hydroxyl group of emodin is engaged in tight H-bonding with Glu205 and Glu206 (Fig. 3A), suggesting a mechanism of binding of emodin to the DPP4 active site. The binding modes showed that emodin was bound to the active site of DPP4 with the hydroxyl moiety but did not form hydrogen bonds with other amino moieties such as Tyr547 or Trp629 (*Ji et al., 2014*; *Kim et al., 2005*), which may affect the activity of emodin. The compounds with similar structure as emodin that have hydroxyl group at similar site

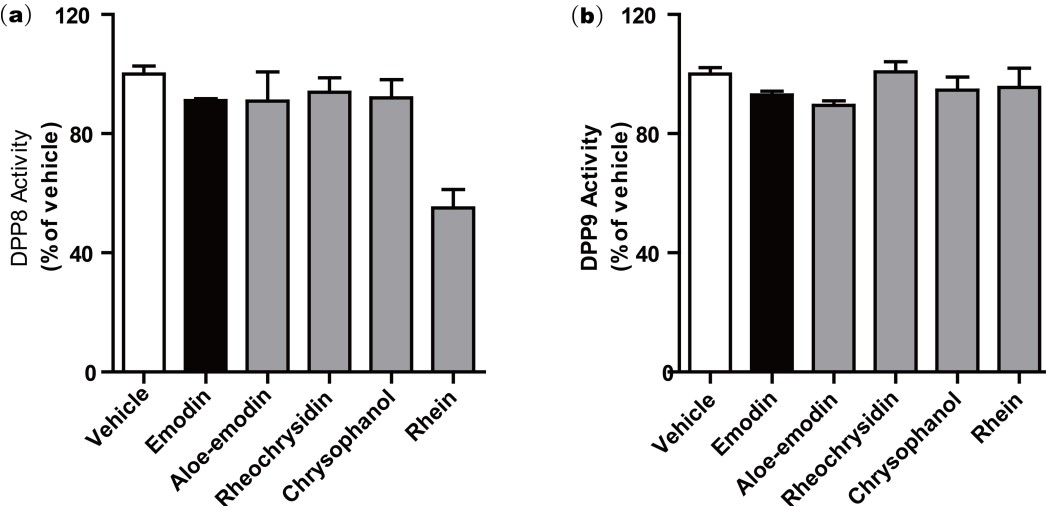

**Figure 2** Anthraquinone compounds do not inhibit either (a) DPP8 or (b) DPP9.

**Table 1** Anthraquinone compounds inhibit DPP4 activity.

| Compound | $R_1$ | $R_2$ | $IC_{50}$ of DPP4 Inhibition (µM) | Ki (µM) | Binding Energy (kcal/mol) |
|---|---|---|---|---|---|
| Emodin | -CH$_3$ | -OH | 5.76 ± 0.42 | 0.85 ±0.06 | −5.19 |
| Aloe-emodin | -H | -CH$_2$OH | 16.02 ±4.24 | 2.37 ±0.62 | −5.31 |
| Rheochrysidin | -CH$_3$ | -OCH$_3$ | >100 | | −4.60 |
| Chrysophanol | -H | -CH$_3$ | >100 | | −4.46 |
| Rhein | -H | -COOH | 23.06 ±3.57 | 3.42 ±0.53 | −4.73 |

**Notes.**
Values (µM) are means ± SE.

could also form H-bond with Glu205 and Glu206, and these compounds (aloe-emodin and rhein) also showed DPP4 inhibitory activity. In comparison, compounds without the hydroxyl group at R2 location (rheochrysidin and chrysophanol) lack the ability to form H-bond with DPP4 at active site, thus they showed weakest DPP4 inhibitory activity. Following dialysis assay suggested emodin binding to the DPP4 active site in a reversible manner (Fig. 3B). To evaluate the DPP4 inhibitory activity of emodin *in vivo*, we orally administered emodin to Balb/c mice.

## Emodin inhibits DPP4 activity *in vivo*

DPP4, also known as adenosine deaminase complexing protein 2 or T-cell activation antigen CD26, is a member of the large family of proteases. DPP4 is associated with immune regulation, signal transduction and apoptosis. Recent reports shown that DPP4

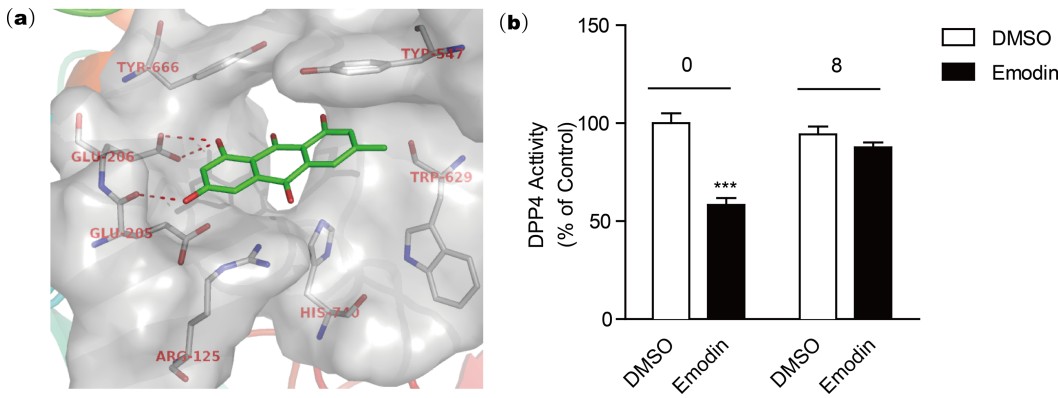

**Figure 3** Docking model reveals the binding mode of emodin to DPP4 protein.

correlates closely to diabetes and cancer. Our labs have focused attention on DPP4 for the screening of inhibitors, such as emodin from **Rheum palmatum** Linn. As a natural product and active ingredient of various Chinese herbs, emodin exerts its anti-diabetic effects partially by upregulating the expression of the pancreas L-type calcium channel in streptozotocin (STZ)-induced dyslipidemic diabetic rats (*Zhao et al., 2009*) and by inhibiting 11 beta-hydroxysteroid dehydrogenase type 1 (11$\beta$-HSD1) activity in diet-induced obese (DIO) mice (*Feng et al., 2010*; *Wang et al., 2012*). *Xue, Ding & Liu (2010)* have also reported that emodin exerts anti-diabetic effects against PPAR-gamma in mice either administered a high-fat diet or treated with low-dose STZ to induce diabetes. *Song et al. (2013)* have reported that emodin regulates glucose homeostasis *in vivo* via AMP-activated kinase (AMPK) activation. Emodin has also been shown to decrease blood glucose in rats with diabetes induced by low-dose STZ combined with high energy intake (*Wu et al., 2014*). These data clearly show that emodin exerts anti-hypoglycemic effects through diverse mechanisms, which is in line with the results of our screening analysis.

To test the inhibitory activity of emodin on DPP4 *in vivo*, an animal experiment was conducted by oral administration of emodin (3, 10 and 30 mg/kg, P.O., $n = 5$) to Balb/c mice, followed by plasma collection at different time points to measure DPP4 activity in the blood. Plasma samples were collected at 0, 0.5, 1, 2 and 4 h after the oral dose of emodin, and plasma DPP4 activity was measured with a DPP4-Glo assay kit. The results suggested that emodin treatment (3, 10 and 30 mg/kg, P.O.) in mice decreased the plasma DPP4 activity in a dose-dependent manner (Fig. 4). The lowest does of emodin (3 mg/kg) decreased plasma DPP4 activity from baseline, although these levels rebounded after 1 h, whereas 10 and 30 mg/kg doses of emodin decreased plasma DPP4 activity and maintained the lower levels until 2 h post-treatment (Fig. 4). This dose-dependent manner is probably due to the pharmacokinetics of emodin, and 10 and 30 mg/kg have maximized the pharmacokinetic coverage, while 3 mg/kg emodin is only sufficient to maintain 1 h DPP4 inhibition *in vivo*. In the subsequent experiment, emodin (30 mg/kg, P.O.) was administered on Balb/C mice ($n = 5$) or ob/ob (-/-) mice ($n = 5$). The DPP4 activity, blood glucoselevels and GLP-1 activity were tested 0, 1, 2, 4, 8 h after administration. The baseline level of DPP4 was higher in ob/ob (-/-) mice compared to Balb/C mice, and the data demonstrated a significant

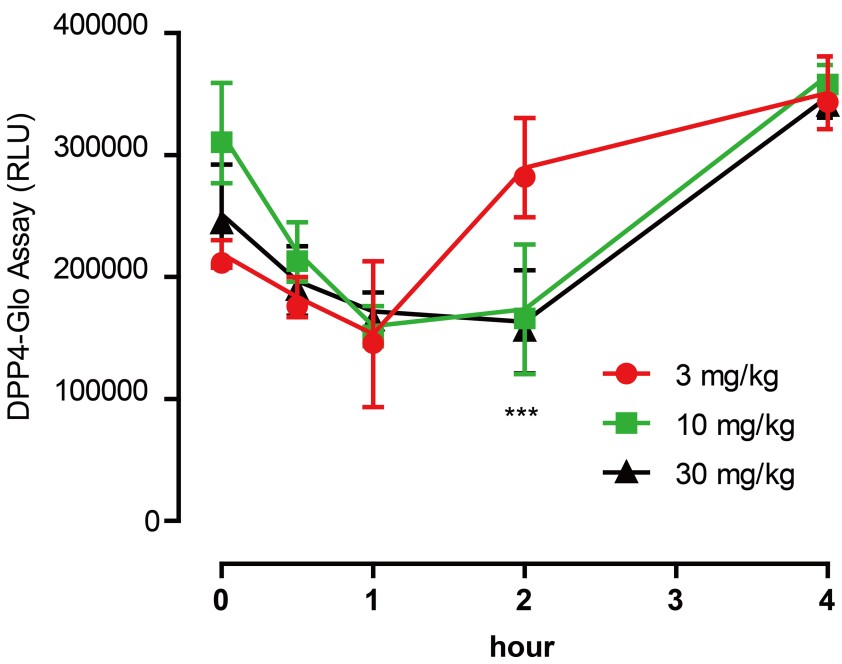

**Figure 4 Emodin treatment (3, 10 and 30 mg/kg, P.O.) in mice decreased the plasma DPP4 activity in a dose-dependent manner.**

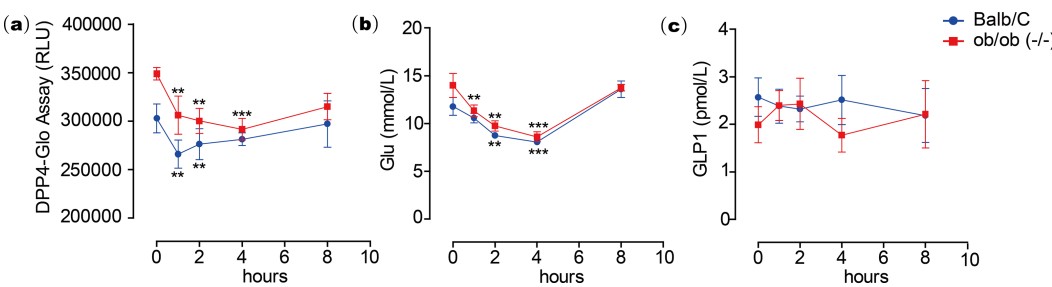

**Figure 5 Emodin treatment in mice.**

downregulation of DPP4 activity after emodin oral administration in both Balb/C mice and ob/ob (-/-) mice (Fig. 5A). This downregulation maintained 2 h in Balb/C mice, and 4 h in ob/ob (-/-) mice, which rebounded afterwards (Fig. 5A). Meanwhile, emodin down-regulated blood glucose level after oral administration. In both Balb/C mice and ob/ob (-/-) mice, emodin treatment significantly downregulated blood glucose levels from baseline (Fig. 5B), and this downregulation maintained for 4 h post treatment, and the blood glucose levels returned to baseline 8 h after treatment. Plasma GLP-1 activity was also measured. GLP-1 has a low baseline activity and emodin showed marginal effect on plasma GLP-1 activity (Fig. 5C). These results demonstrate that emodin inhibits DPP4 activity *in vivo*, which may contribute to its anti-diabetic properties.

Emodin has been detected in various Chinese herbs and is efficacious against inflammatory disorders and cancer (*Shrimali et al., 2013*; *Wei et al., 2013*) and liver cirrhosis

(*Woo et al., 2002*); furthermore, emodin has demonstrated immunosuppressive (*Kuo et al., 2001*) and antibacterial (*Wang & Chung, 1997*) properties. Although many studies have shown the effects of emodin on metabolic abnormalities (especially diabetes), the molecular mechanisms involved have not been thoroughly studied. Our study shows for the first time that emodin is a selective DPP4 inhibitor both *in vitro* and *in vivo* which may explain the anti-diabetes effects of this compound.

The toxicity of emodin should also be paid attention to and it has been reported in the previous publications (*National Toxicology Program, 2001*; *Wang & Chung, 1997*). In these reports, there was no evidence of carcinogenic activity of emodin either in male F344/N rats or female B6C3F1 mice. Although emodin exposure resulted in increased incidences of renal tubule pigmentation in male and female mice and increased incidences of nephropathy in female mice, the emodin doses used in these reports (280–2,500 ppm) were much higher than the emodin dose in our reports. Our highest *in vivo* dose 30 mg/kg (equivalent to 30 ppm) is almost one ninth of the lowest dose used in these reports. However, this brings an attention to the chronic toxicity of emodin in the treatment of diabetes in the future. On the other hand, some reports have addressed that emodin isolated from rhubarb may have anti-cancer effects on a few human cancers (*National Toxicology Program, 2001*).

## CONCLUSIONS

DPP4 is a well-characterized therapeutic target for type II diabetes treatment, and there have been extensive drug discovery activities reported in this area. However, very few literature has reported natural compounds with activity against DPP4 (*Fan et al., 2013*). The current study was the first to screen a natural compound library consisting of Chinese herbal medicines and marine organisms, with the goal of identifying novel small molecules that inhibit DPP4. As a result, we discovered that emodin, a compound belonging to the anthraquinone family, selectively inhibited *in vitro* DPP4 activity with an IC50 of 5.7 μM.

To further understand the binding mechanism of emodin and DPP4, we conducted a molecular docking model by simulating the emodin binding mode at the DPP4 active site. The docking assay revealed that emodin interacts with the DPP4 active site and forms H-bonds with Glu205 and Glu206 at the active site of DPP4.

Based on the *in vitro* data and the docking model, we subsequently conducted animal experiment by orally administering emodin to Balb/C mice and ob/ob (-/-) mice. Plasma DPP4 activity was inhibited by emodin administration in a dose-dependent manner, and the blood glucose levels were decreased in both mice strains.

Together, these results suggest that emodin is a small molecule inhibitor of DPP4, showing activity both *in vitro* and *in vivo*. Emodin, as a novel anti-hypoglycemic compound, may stimulate new drug discovery for the treatment of type 2 diabetes.

### Funding

This project was sponsored by the Scientific Research Foundation of the Third Institute of Oceanography, SOA (No. 2015004). This work was also supported by Regional Demonstration of Marine Economy Innovative Development Project (No. 16PYY005SF09). The funders had no role in study design, data collection and analysis, decision to publish, or preparation of the manuscript.

### Grant Disclosures

The following grant information was disclosed by the authors:
Scientific Research Foundation of the Third Institute of Oceanography, SOA: 2015004.
Regional Demonstration of Marine Economy Innovative Development Project: 16PYY005SF09.

### Competing Interests

The authors declare there are no competing interests.

### Author Contributions

- Zhaokai Wang conceived and designed the experiments, contributed reagents/materials/analysis tools, prepared figures and/or tables, reviewed drafts of the paper.
- Longhe Yang conceived and designed the experiments, contributed reagents/materials/analysis tools.
- Hu Fan and Peng Wu performed the experiments.
- Fang Zhang and Chao Zhang analyzed the data.
- Wenjie Liu and Min Li conceived and designed the experiments, wrote the paper, reviewed drafts of the paper.

### Animal Ethics

The following information was supplied relating to ethical approvals (i.e., approving body and any reference numbers):

Animal Welfare Committee of Research Organization (X201352) of Xiamen University.

### Data Availability

The raw data has been supplied as a Supplementary File.

### Supplemental Information

Supplemental information for this article can be found online at http://dx.doi.org/10.7717/peerj.3283#supplemental-information.

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
