# Peer review of "Screening of a natural compound library identifies emodin, a natural compound from Rheum palmatum Linn that inhibits DPP4"

_PeerJ, doi:10.7717/peerj.3283_

## Round 0.1 · original submission · Major Revisions

Please address the comments of our reviewers.

Personal review-level comment by the editor: In the analysis of the binding mode you may want to highlight more explicitly the importance of the H-bond between the emodin OH group and the C=O of Glu205, since the compounds which lack the ability to for such a bond (rheochrysidn and chrysophanol) are also the ones with the weakest binding ability.

Reviewer 1 ·

Basic reporting

no comment

Experimental design

no comment

Validity of the findings

no comment

Additional comments

Wang and Colleagues investigated screening of a natural compound library and emodin from Rheum palmatum Linn inhibits dipeptidyl peptidase 4 (DPP 4) activity. The authors found that DPP 4 activity was reduced by emodin in vitro and in vivo. In additoin, DPP 8 and 9 were not inhibited by emodin in vitro. The authors conclude that emodin may stimulate new drug discovery for the type 2 diabetes. There are some comments to the authors for improvement of the manuscript.

1) Please describe the purification method of emodin from Rheum palmatum Linn in the Materials and Methods section.

2) Line 101: The authors should describe the animal data of number, sex and age.

3) Line 119-120: It would be useful to provide data for 155 naturally compound names and DPP 4 activitis (% of vehicle control) in supplementary Table.

4) Line 126-128 and 136-138: The authors should show the Ki value and Binding energy (kcal/mol) of emodin, aloe-emodin, rhein, rheochrysidin and chrysophanol, respectively. It would be useful to provide the data in Table 1.

5) Line 142-144: It seems that rhein inhibits DPP 8 activity in Fig. 2a. The authors should explain and discuss for this point.

6) The authors should show the alteration of blood glucose levels, active GLP-1 and GIP circulating levels, blood insulin levels in emodin-administered mice.

7) The conclusion is too long. Please summarize briefly.

8) Line 204-205: The authors should check the literature more carefully. For example, Fan et al (Evid. Based. Complement. Alternat. Med. 2013 (2013) 479505.) have shown DPP IV inhibitors from natural products.

Reviewer 2 ·

Basic reporting

Clear and unambiguous, professional English used throughout.
Literature references, sufficient field background/context provided.
Professional article structure, figs, tables. Raw data shared.
Additional data be required.

Experimental design

Original primary research within Aims and Scope of the journal.
Research question well defined, relevant & meaningful.
Rigorous investigation performed to a high technical & ethical standard.
Methods described with sufficient detail & information to replicate.

Validity of the findings

There are some effects and novelty, but also need additional evidence to prove the conclusion.
In this manuscript, the authors described the emodin inhibits DPP4 activity and may represent a novel therapeutic for the treatment of type 2 diabetes.
Although several DPP 4 inhibitors have been approved for the treatment of T2DM, authors reported emodin of natural compounds can inhibit DPP4 activity. However, there have been many studies reported that emodin exerts its anti-diabetic effects partially by upregulating the expression of the pancreas L-type calcium channel, or by inhibiting11 beta-hydroxysteroid dehydrogenase type 1 activity, or against PPAR-gamma, or emodin regulates glucose homeostasis in vivo via AMP-activated kinase activation【In lines 161-169 of this manuscript】. The novelty of this study is that elucidate the molecular mechanism of emodin in inhibition of DDP4 activity.

Additional comments

In this manuscript, the authors described the emodin inhibits DPP4 activity and may represent a novel therapeutic for the treatment of type 2 diabetes.
Although several DPP 4 inhibitors have been approved for the treatment of T2DM, authors reported emodin of natural compounds can inhibit DPP4 activity. However, there have been many studies reported that emodin exerts its anti-diabetic effects partially by upregulating the expression of the pancreas L-type calcium channel, or by inhibiting11 beta-hydroxysteroid dehydrogenase type 1 activity, or against PPAR-gamma, or emodin regulates glucose homeostasis in vivo via AMP-activated kinase activation【In lines 161-169 of this manuscript】. The novelty of this study is that elucidate the molecular mechanism of emodin in inhibition of DDP4 activity.

Major comments
1, In the manuscript , there is no direct evidence prove that relationship between emodin inhibiting DPP4 activity and treatment of type 2 diabetes in this study, if you can provide evidence that emodin binds to DPP4 in vivo or in vitro. Figure 3 predicts the interaction between emodin and DPP4, and the subsequent data present the inhibitory result of DPP4 activity. However, there have no evidences showing direct binding of emodin and DPP4. Could author supply the binding results. For example, treat the PVDF membrane, in which the wild DPP4 and its active site of DPP4 mutant proteins have been immobilized, with Fluorescent labeling of emodin (or other Immunofluorescence staining methods) and detect the fluorescence of membrane. If Cinchonine could bind to wild DPP4, the fluorescence will differ between wild DPP4 and its mutant.

2, To test the inhibitory activity of emodin on DPP4 in vivo, an animal experiment was conducted by oral administration of emodin (3, 10 and 30 mg/kg, P.O.) to Balb/c mice, followed by plasma collection at different time points to measure DPP4 activity in the blood. At the same time, the activity of glucagon-like peptide 1 in the plasma should be compared before and after emodin treatment.


3, In the determination of DDP4 activity, there is no positive control group? (Figures 1, 4, Table 1). In addition, in Fig. 4,although the higher doses (10 and 30 mg/kg) of emodin decreased plasma DPP4 activity, but this decrease was maintained just 2 hours post-treatment, instead, showed increasing tendency from 4 hours, how about the DPP4 activity after emodin treatment 6, 8 hours ? Please add the comparison data with the positive control? And please explain that if the role of emodin is only 2 hours, how do to apply in clinical?

4, What is the concentration of emodin used in previous studies【In lines 161-169 of this manuscript】? In this study, how determine the concentration of emodin?


5, The effect of 3, 10, or 30 mg / kg of emodin on blood glucose should be shown.

---

## Round 0.2 · Minor Revisions

Some of the information metnione in your response has not been included in the manuscript. These are:

a) The binding energies are still missing from table 1.
b) The number, age, and sex of the animals is also missing from the Materials and Methods section.
c) Origin of emodin, chrysophanol, etc. . I noticed, too, that you state "Marine-derived compounds were isolated and purified from marine organisms in our lab." but do not state which compounds you are referring too nor provide references to the purification protocols.

Reviewer 1 ·

Basic reporting

no comment

Experimental design

no comment

Validity of the findings

no comment

---

## Round 0.3 · accepted · Accept

Thank you for addressing the last remaining issues.